# The Multifunctional Role of KCNE2: From Cardiac Arrhythmia to Multisystem Disorders

**DOI:** 10.3390/cells13171409

**Published:** 2024-08-23

**Authors:** Ming Song, Yixin Zhuge, Yuqi Tu, Jie Liu, Wenjuan Liu

**Affiliations:** Department of Pathophysiology, Medical School, Shenzhen University, Shenzhen 518060, China; 2200243027@email.szu.edu.cn (M.S.); 2210245084@email.szu.edu.cn (Y.Z.); 2300243031@email.szu.edu.cn (Y.T.)

**Keywords:** KCNE2, potassium channel, cardiovascular diseases, neurological diseases, hypothyroidism, type 2 diabetes mellitus

## Abstract

The KCNE2 protein is encoded by the *kcne2* gene and is a member of the KCNE protein family, also known as the MinK-related protein 1 (MiRP1). It is mostly present in the epicardium of the heart and gastric mucosa, and it is also found in the thyroid, pancreatic islets, liver and lung, among other locations, to a lesser extent. It is involved in numerous physiological processes because of its ubiquitous expression and partnering promiscuity, including the modulation of voltage-dependent potassium and calcium channels involved in cardiac action potential repolarization, and regulation of secretory processes in multiple epithelia, such as gastric acid secretion, thyroid hormone synthesis, generation and secretion of cerebrospinal fluid. Mutations in the KCNE2 gene or aberrant expression of the protein may play a critical role in cardiovascular, neurological, metabolic and multisystem disorders. This article provides an overview of the advancements made in understanding the physiological functions in organismal homeostasis and the pathophysiological consequences of KCNE2 in multisystem diseases.

## 1. Introduction

The KCNE gene family comprises a cluster of genetically related genes that encode products with similar structures and functions. To date, there have been a total of five genes associated with KCNE, namely KCNE1–5. These genes are responsible for producing protein products known as Mink proteins and Mink-related peptides 1–4 (Mink-related peptides 1–5, MiRPs) [1]. These small proteins, known as auxiliary β-subunits, are unable to form ion channels on their own. However, they play a role in regulating the α-subunits of various potassium channel proteins, including KCNQ1, human ether-a-go-go-related gene (HERG), Kv2.1, Kv4.2, Kv4.3 and HCN. When combined with these α-subunits, they form a complex that functions as a voltage-dependent potassium channel. The KCNE family exerts its influence on voltage-dependent potassium channels through a variety of mechanisms, including modulation of gating kinetics, drug sensitivity, alteration of channel proteins and transport [2]. Abnormalities in KCNE family members, whether structural or functional, can either directly or indirectly affect potassium channel currents and result in disruptions in the electrical activity of cardiomyocytes. This, in turn, may lead to the development of different types of arrhythmias [3].

The initial member of the KCNE protein family to be identified was KCNE1, also known as the Mink protein, which was first documented by Murai in 1989. Mutations in KCNE1, which encodes the β-subunit of the slow-activating delayed rectifier potassium channel (I_Ks_) in cardiomyocytes, are associated with long QT syndrome 5 (LQT5) [4,5,6]. The discovery of KCNE2 was made by Abbott and colleagues, a decade after the initial discovery of KCNE1 [7]. The KCNE2, also known as the Mink-related peptide 1 (MiRP1), exhibits a broad distribution across multiple organs, with significant expression levels observed in the brain, heart, skeletal muscle, pancreas, placenta and kidney. KCNE2 has been found to exert a significant impact on the electrical properties of various cell types and tissues through the modulation of different potassium channel α-subunits, including HERG [8] (fast component of delayed rectifier potassium current), Kv4.x (x = 2/3) [9,10] and Kv3.4 (transient outward current, I_to_), KCNQ1 (slow component of delayed rectifier potassium current) and HCNx (hyperpolarization-activated cyclic nucleotide-gated channels) [11,12,13].

The initial discovery was that KCNE2 is essential for the maintenance of cardiac electrophysiological stability. Scientific studies have demonstrated that mutations in the KCNE2 gene or aberrant protein expression can give rise to long QT syndrome (LQTS) [8]. The role of KCNE2 in maintaining cardiac electrical stability is further supported by the evidence that genetic variations and alterations in KCNE2 result in acquired LQT6 [7].

As the studies have progressed, it has become clear that KCNE2 has a more extensive function than just the maintenance of cardiac electrical stability (Figure 1). KCNE2 may also play a regulatory role in cardiac structure and function. KCNE2 deficiency causes cardiac dysfunction, leading to right heart failure and the development of liver fibrosis [14,15,16]. KCNE2 deletion has also been documented to cause sudden cardiac death [15]. KCNE2 is highly expressed in ventricular and Purkinje tissues and widely distributed throughout the body, including the stomach, brain, heart, skeletal muscle, pancreas, kidney, placenta and other tissues and organs. Notably, KCNE2 expression is highest in the stomach [17]. The knockdown of KCNE2 in mice disrupts gastric acid secretion. The reduction in KCNE2 also results in an increase in the level of angiotensin II in the blood and dysfunction of the adrenal glands [15]. In addition, a lack of KCNE2 can lead to dyslipidemia, diabetes and anemia, which in turn can contribute to hyperkalemia and prolongation of the QTC interval of the action potential [18]. In 2017, Abbott discovered that KCNE2 is abundant in the tissues of lactating animals and is associated with several diseases in humans and mice. They also observed that lack of KCNE2 led to the development of type 2 diabetes mellitus (T2DM) in mice [19]. This review summarizes the molecular characteristics of KCNE2, including its molecular structure and tissue-specific distribution. We focus on the multifaceted functions of KCNE2, with a particular emphasis on its roles in the development of various pathological conditions, including cardiovascular, neurological, metabolic and multisystem disorders.

## 2. Molecular Characterizations

### 2.1. Structure

The KCNE gene family comprises a cluster of genes that have similar structural features and produce products that exhibit both structural and functional similarities. To date, five distinct KCNE-related genes have been identified in the human population, with the designation of KCNE1–KCNE5. The protein products have been designated as Mink-related peptides (Mink-related peptides, MiRPs), or alternatively, KCNE proteins. These proteins are composed of three components: the N-terminal extracellular region, the transmembrane region comprising only a single transmembrane fragment [20] and the C-terminal cytoplasmic portion.

The KCNE2 gene is responsible for the encoding of the protein MiRP1, also known as KCNE2. This protein functions as the β-subunit of various voltage-dependent potassium channels [21,22], including KCNQ1, Kv4.x, HCN and others. The protein comprises 123 amino acid residues and contains two N-glycosylation sites, along with two phosphorylation sites that undergo protein kinase C-mediated phosphorylation (thr71 and ser74) [23].

The KCNE genes show minimal homology or similarity, although they do share a transmembrane (TM) span and a proven or potential PKC phosphorylation site in the proximal region of the intracellular membrane [24] (Figure 2). KCNE2 is thought to have arisen from a gene duplication event, as it is located on chromosome 21q22.1, approximately 79 kb from KCNE1. The open reading frames of KCNE1 and KCNE2 share 34% identity [7].

### 2.2. Tissue-Specific Distribution

KCNE2 has a ubiquitous distribution in all tissues and organs of the body, with notable expression in the stomach, brain, heart and other tissues and organs. In particular, the highest level of KCNE2 expression is observed in the stomach and choroid plexus [17].

KCNE2 is unevenly expressed in different parts of the heart. The research findings suggest that the KCNE2 protein is more abundant in the ventricles and Purkinje tissues compared to the atria. KCNE2 has also been found to be expressed in the mural cells of the stomach [25]. The KCNQ1–KCNE2 channel has been identified as a significant channel for the K^+^ recycling current in the apical parietal cell, which is essential for the secretion of gastric acid [26].

Meanwhile, the KCNE2 protein is highly expressed in the apical membrane of the choroid plexus epithelium (CPe) [27]. Similar to parietal cells, KCNE2 combines with KCNQ1 to form a K^+^ channel on the apical aspect of CPe, which plays a role in regulating the baseline cerebrospinal fluid (CSF) [28].

In addition to these findings, KCNE2 may also be co-expressed with KCNQ1 in the basolateral region of thyroid cells [15]. The expression of the KCNQ1–KCNE2 channel is upregulated in response to an increased requirement for thyroid hormone synthesis, which is likely to enhance the transport capacity of sodium/iodide symporter (NIS), which is situated on the basolateral side of thyroid cells. It facilitates the active transportation of iodide, a crucial constituent of thyroid hormone, from the circulation into thyroid cells [29].

### 2.3. Transcription Mechanisms

It is now well established that alterations in KCNE2 expression levels can lead to both physiological processes and pathological disorders. Regrettably, there is a paucity of research on the regulatory processes governing the control of KCNE2 expression. While the transcriptional start sites (TSSs) of the human KCNE2 gene have been identified, the 5′-flanking sequences of TSSs do not exhibit any discernible promoter activity [30]. Consequently, the promoter regulatory region of the KCNE2 gene in any species has yet to be fully characterized.

Evidence has shown that the cardiac KCNE2 gene—the subunit, which assists in the functioning of ion channels—is directly regulated by 17-β estradiol (E2). The expression of cardiac KCNE2 transcripts was observed to be increased in animals that were stimulated with 17-β estradiol (E2) [31].

It has been observed that KCNE2 is less active in the failing human ventricles compared to the non-failing human hearts [23,32]. However, the mechanism is unclear and needs to be further investigated.

## 3. Pluripotent Biological Effects of KCNE2

### 3.1. KCNE2 Modulates the Ca^2+^ Channels

Voltage-gated L-type Ca^2+^ channels (LTCCs) play a vital role in the heart. The entry of Ca^2+^ through LCCs stimulates the activation of Ca^2+^ release channels in the sarcoplasmic reticulum (SR) via a mechanism known as Ca^2+^-induced Ca^2+^ release (CICR). This leads to the release of Ca^2+^ from ryanodine receptors, which in turn initiates muscle contraction [33].

KCNE2 affects the opening and closing properties of LTCC in cardiomyocytes. KCNE2 overexpression causes a positive change in the voltage required for activation and a negative change in the voltage required for inactivation of LTCC. In addition, KCNE2 slows down the recovery of LTCC from inactivation and accelerates the process of deactivation. These findings suggest that the changes in the opening and closing properties of LTCC, influenced by KCNE2, may partially explain how KCNE2 regulates LTCC in cardiomyocytes. Furthermore, overexpression of KCNE2 reduced the density of L-type Ca^2+^ current (I_Ca,L_). An opposite result was observed when KCNE2 was knocked down [34].

### 3.2. KCNE2 Modulates the HCN Channels

H. Yu and colleagues discovered that KCNE2 is the only member of the KCNE family known to affect the functionality of HCN subunits. No functional effects of KCNE1 were observed on either HCN1 or HCN2 currents in oocytes [35]. The expression pattern of KCNE2 in cardiac tissues closely resembles that of HCN channels, with the highest level of expression observed in the sinoatrial node, followed by the conduction tissues and the atria. Jihong Qu et al. demonstrated that when KCNE2 was co-expressed with HCN1 or HCN2 in Xenopus laevis oocytes, it led to increased and faster activation currents compared to expressing either HCN isoform alone [36]. Nevertheless, KCNE2 did not affect the activation midpoint of either isoform. Concurrently, studies have demonstrated the interaction between KCNE2 and HCN4 isoforms [37]. Co-expression of KCNE2 increases the amplitude of HCN4 current. However, it has also been shown that the presence of KCNE2 further slows down the activation kinetics. KCNE2 and HCN2 have a shared function of co-assembly in cardiac cells, resulting in an increased expression amplitude of pacemaker currents and faster activation and deactivation kinetics [38].

### 3.3. KCNE2 Modulates the HERG Channels

GW Abbott et al. found that KCNE2 accelerated the degradation of HERG protein and diminished its presence throughout the cell [7]. As a result, the amount of HERG protein on the cell surface decreased, and the amplitude of HERG current was inhibited. Phosphorylation of residue S98 was found to be necessary for the inhibitory effects of KCNE2 on both the amplitude of HERG current and the amount of HERG protein [23]. Experimental studies have demonstrated the crucial role of phosphorylation at S98 in the inhibitory effects exerted by KCNE2 on the amplitude and expression levels of HERG channels, as well as the location of S98 within the cytoplasmic structural domain of KCNE2. The phosphorylation status of KCNE2 may influence the interaction between vesicles carrying KCNE2 and other cytoplasmic proteins involved in the recognition of cargo and the subsequent transportation of these vesicles [39].

### 3.4. KCNE2 Modulate the I_to,fast_ Channel

The transient outward current (I_to_) initiates the early repolarization phase of the action potential and contribute importantly to plateau potentials and the whole repolarization phase by regulating Ca^2+^ and other K^+^ currents [40,41]. The rapidly recovering I_to,fast_ and slowly recovering I_to,slow_ components are differentially expressed in the myocardium. Distinct pore-forming (alpha) subunits underlie the two I_to,fast_ components: Kv4.2/Kv4.3 subunits encode I_to,fast_, whereas Kv1.4 encodes I_to,slow_ [42,43,44,45].

Co-expression of KCNE2 with Kv4.2/Kv4.3 in Xenopus oocytes was observed to significantly decelerate the process of I_to,fast_ activation and deactivation, while also shifting the voltage dependence of activation to a positive membrane potential [10,46,47]. However, there was a discrepancy between the result observed for KCNE2 and the anticipated outcome in the CHO cell translation system.

Erich Wettwer and colleagues proposed that KCNE2 could play a crucial role in the natural I_to,fast_ channel complex, particularly in human epicardial cardiomyocytes [10]. Furthermore, it was determined that KCNE2 exhibits unique characteristics, as its co-expression most precisely reproduces the distinctive “overshoot” phenomenon observed following the reactivation of I_to,fast_ channel inactivation in the epicardium of the human left ventricle [47]. KCNE2 forms complexes with Kv4.2 in the ventricles of adult mice, resulting in increased currents densities and slows its time-to-peak and inactivation [16].

WJ Liu et al. found that KCNE2 controls the opening and closing of the I_to,fast_ channel in natural cardiomyocytes. KCNE2 knockdown consistently increased the gating kinetics in both neonatal and adult cardiomyocytes. Overexpression of KCNE2 reduced the activation and inactivation of I_to,fast_ in neonatal cardiomyocytes. However, it had no effect on the gating properties of I_to,fast_ in adult cardiomyocytes, which may be due to the saturation of KCNE2 [48] (Table 1).

### 3.5. KCNE2 Modulates the KCNQ1

#### 3.5.1. KCNQ1 in Stomach

KCNE2 confers to KCNQ1 the ability to be activated by low extracellular pH, whereas KCNQ1 lacking any KCNE is suppressed by external protons. KCNQ1-KCNE2 acts as a channel for potassium cycling in the parietal cells, allowing potassium ions to be returned to the gastric lumen in exchange for protons (Figure 3). This process contributes to the acidification of the stomach [49,50,51]. In the absence of KCNE2, KCNQ1 moves erratically to the basolateral side of the cell, resulting in a failure to replace potassium (K^+^) in the gastric lumen.

#### 3.5.2. KCNQ1 in Thyroid

KCNQ1 and KCNE2 have been reported to combine to form a K^+^-channel in thyrocytes that is activated by thyrotropin. The KCNQ1-KCNE2 channels are located in the basolateral region of thyroid epithelial cells, and the expression of KCNQ1 and KCNE2 proteins is increased by the thyrotropic hormone (TSH) and is always active. This K^+^-channel is necessary for the proper production of the thyroid hormone [15]. KCNQ1–KCNE2 channels play a specialized role in facilitating the uptake of iodinated thyroid via the sodium/iodide isotropic transporter (NIS) [52] rather than being involved in the organization of iodide after uptake.

#### 3.5.3. KCNQ1 in Choroid Plexus Epithelium (CPe)

The CPe is a specialized, non-responsive layer of epithelial cells that is found in the lateral and fourth ventricles of the brain. It plays a crucial role in the production, secretion, and regulation of the CSF [53]. Therefore, the regulation of ion flow from the basolateral (blood) side of the CPe to the apical (CSF) side is crucial, as it has a significant impact on several physiological features of the central nervous system, such as neuronal function and intracranial pressure [54,55].

KCNE2 is highly expressed in the apical membrane of the CPe. Similar to mural cells, KCNE2 combines with KCNQ1 to form apical potassium channels at the CPe [27]. TK Roepke et al. demonstrated that the disruption of KCNE2 did not result in any alteration to the baseline CSF [K^+^], but did in fact increase the outward K^+^ current. This suggests that the KCNQ1–KCNE2 channel probably does not play a significant role in the regulation of the baseline CSF [K^+^], and instead provides a K^+^ efflux pathway which acts to counteract the K+ influx through the apical Na^+^/K^+^ ATPase. It is conceivable that KCNQ1-KCNE2 channels are indispensable for regulating CSF [K^+^] in instances of acute disruption. KCNE2 deletion results in KCNQ1 trafficking to the CPe basolateral membrane, whereas KCNQ1 rerouting may be a pivotal factor in CPe dysfunction.

Additionally, TK Roepke et al. discovered that KCNE2 also forms apical potassium channels at the CPe with Kv1.3 (KCNA3), contributing to apical K^+^ efflux [27]. The researchers demonstrated the functional impact of co-expressing mouse KCNE2 and rat Kv1.3 in CHO cells. KCNE2 was observed to inhibit 60% of KCNA3 currents, providing the first evidence of the modulatory effect of KCNE2 on Kv1.3, which was also observed in the CPe of KCNE2 knockout mice [27]. Kv1.3 preserves its voltage-dependent functionality when co-expressed with KCNE2, whereas KCNQ1 loses its voltage-dependence when co-expressed with KCNE2.

Indeed, previous research conducted by GW Abbott et al. and Morten Grunnet et al. investigated the potential interaction between KCNE2 subunits and Kv1.3 in Xenopus oocyte expression system [7,56]. No observable impact of KCNE2 on Kv1.3 channels expressed in Xenopus oocytes. In another study, L Sole et al. observed that human KCNE2 did not modulate rat Kv1.3 function in HEK cells [57].

The study reveals that the precise cause for this discrepancy remains unclear. It is conceivable, however, that this interaction may exhibit cell type- and/or KCNE2 species-specific dependence, as observed with some other KCNE2-α subunit partnerships.

### 3.6. The Effect of KCNE2 on the Visual Pathway

KCNE2 is found predominantly in the outer plexiform layer (OPL), where it is thought to be present at the initial synapse of the cone visual pathway and potentially in rod bipolar cells (BCs). Moritz Lindner et al. observed the presence of KCNE2 immunoreactivity in cone cells in mice. Several ion channels that interact with KCNE2 are also present in the cone cell outer segments, including Kv2.1and HCN1 [58]. The authors hypothesize a potential regulation of KCNE2 on Kv2.1 in cone cells.

Kv2.1 and Kv8.2 combine to form a heterodimer in both cone and rod cells [58]. A loss-of-function mutation in Kv8.2 can lead to cone dystrophy with a supernormal rod response (CDSRR) due to an increased potassium conductance (I_K,x_) [59,60], a genetic retinal disorder. However, CDSRR is characterized by a specific early deterioration of cone cells, while rod cells show greater adaptability [60]. Moritz Lindner et al. propose that the expression of KCNE2 regulates Kv2.1 in cone cells, while the functional loss of Kv8.2 in CDSRR has a greater impact on I_K,x_, hence promoting the susceptibility of cone cells [61]. This expression pattern has been identified not only in mice but also in other animals that have evolved over a period of 300 million years. It is therefore highly likely that it is conserved in all vertebrates. The localization of KCNE2 in the OPL is also conserved in primates. It is thus very likely that this is also the case in humans [61].

## 4. KCNE2 and Diseases

### 4.1. Cardiovascular Diseases

#### 4.1.1. Cardiac Arrhythmia

Mutations in KCNE2 have associated with hereditary and drug-induced long QT syndrome (LQTS). LQTS is a collection of genetically diverse disorders characterized by arrhythmias, QT interval prolongation and an increased risk of sudden cardiac death [62,63]. Prolongation of the QT interval is a characteristic feature of LQTS, which can occur through two different mechanisms: reduced potassium currents during phase 3 of the action potential (loss of function) or more delayed entry of sodium or calcium ions into the heart muscle cells (gain of function). It can occur in two ways: either by a reduction in the outward potassium current during phase 3 of the action potential (loss of function), or by an increase in the delayed entry of sodium or calcium ions into the cardiomyocyte (gain of function) [64,65]. Three genetic variations in the *Kcne2* gene (Q9E, M54T, and I57T) result in loss of the fast-rectifying delayed potassium currents (I_Kr_) function and are associated with acquired LQTS [7,66].

Clinical LQTS genetic testing in North America through PGx Health’s clinical test called the FAMILION^®^ LQTS test, has provided a comprehensive mutation analysis of KCNE2, the LQT6 susceptibility gene [67]. Contrary to its low prevalence, with an estimated overall prevalence of ≈0.0005%, many KCNE2 variants implicated in LQT6 have higher than anticipated frequencies within population-based exome cohorts. Scheinman MM et al. suggest that loss-of-function KCNE2 rare variants may not be sufficient in isolation to cause LQTS. However, they may confer proarrhythmic susceptibility when triggered by additional environmental or genetic factors. Avoidance of secondary stressors associated with QT prolongation should be the primary focus of clinical management of individuals with loss-of-function KCNE2 variants and normal phenotypes [68].

Additionally, R27C, M23L and I57T mutations of KCNE2 are associated with familial and early-onset atrial fibrillation [66,69]. Recent research has demonstrated that mutations in KCNE2, namely T8A and Q9E, can lead to long QT syndrome (LQTS) [51,66,70,71,72]. Their impact on HERG–KCNE2 channels is characterized by a loss-of-function, leading to an extension of the action potential length and QT interval [7]. Unlike all other KCNE2 mutations associated with LQTS, the KCNE2 R27C mutation resulted in an increase in KCNQ1–KCNE2 current. Nevertheless, the mutation had no impact on the HERG-KCNE2 current [69]. KCNE2 has the ability to interact with other pore-forming subunits, specifically the HCN channel family [35]. Yiqing Yang and colleagues conducted a study to examine the impact of the R27C mutation on the expression of the KCNE2 subunit in HCN1, HCN2 and HCN4. The results of the study revealed that the mutation did not have a significant effect. Yiqing Yang et al. believe that the KCNE2 R27C mutation is unlikely to be the cause of AF through the HCN channel family. The KCNE2 R27C mutation is a newly discovered mutation in familial AF that enhances the activity of potassium currents. Gaining insight into the molecular mechanism of familial AF may also enhance our understanding of the more prevalent causes of AF and potentially pave the way for novel therapeutic approaches [69].

The I57T and M54T mutations of KCNE2 result in significant functional enhancement of the human I_to_ current [73]. Wu et al. investigated the I57T and M54T variants of KCNE2, both of which significantly increased the peak density, slowed down the decay, and accelerated the recovery from inactivation of I_to_ [73].

#### 4.1.2. Heart Failure

Heart failure (HF) is a severe and progressively prevalent condition. It leads to the gradual deterioration of the heart muscle, causing it to lose its ability to effectively pump an adequate amount of blood into the aorta. Consequently, blood accumulates in the heart. Atrial fibrillation is primarily caused by coronary artery disease, with other contributing factors including hypertension, heart valve abnormalities, cardiomyopathy, myocarditis, and arrhythmias [74].

A deficiency of KCNE2, whether cardiac-specific or systemic, leads to dilated cardiomyopathy (DCM) and advanced HF. Ulrike Lisewski and her colleagues discovered that the absence of KCNE2 specifically in the heart has more severe consequences compared to the absence of KCNE2 in the whole body [75]. KCNE2 is not only recognized as an auxiliary subunit of Kv channels, but it also plays a role in regulating the cardiac Cav1.2 [34]. Previous studies have demonstrated that manipulating Cav1.2 genetically leads to the development of fatal DCM in mice. Ulrike Lisewski et al. discovered that the deletion of KCNE2, both globally and specifically in the heart, resulted in an increase in the peak of LTCCs current in the ventricular myocytes of young mice aged 3–6 months. The reversal of this scenario occurred in global KCNE2 null mice at the age of 12–15 months [75]. In addition, the deletion of KCNE2, whether specific to the heart or affecting the entire body, causes a decrease in the inactivation of LTCCs. This decrease is not dependent on age and may lead to an increase in the buildup of cytosolic Ca^2+^ or a longer time required to interrupt other processes. Additionally, it causes a negative shift in the activation of LTCCs, which is dependent on voltage. Therefore, the lack of some elements of Cav1.2 control in mice with a specific absence of KCNE2 in the heart may play a role in the early fatal development of dilated cardiomyopathy and heart failure [75].

Reduced expression of KCNE2 has also been observed in heart failure following myocardial infarction [76]. Reduced KCNE2 plasma channel function is associated with ventricular tachyarrhythmias and Torsades de Pointe (TdP), one of the most important causes of death in patients with heart failure. Nattel S et al. observed a downregulation of major potassium channels in heart failure and myocardial infarction [77]. Alterations in potassium channels and Na^+^/Ca^2+^ exchangers (NCX) may decrease the repolarization reserve and increase the inward current, leading to arrhythmogenesis in the failing heart [78]. Po-Cheng Chang et al. found that valsartan/sacubitril (LCZ696) treatment increased the expression of potassium channels, including ERG, KCNE1 and KCNE2, in a rat MI-HF model [76]. These ion channels are responsible for I_Kr_ and I_Ks_ [78]. Furthermore, it has been observed that KCNE2 is less active in the failing human ventricles as compared to the non-failing human hearts [23,32].

#### 4.1.3. Sudden Cardiac Death

Sudden cardiac death (SCD) is the primary cause of mortality on a global scale. SCD is typically the result of myocardial ischemia-reperfusion (IR) injury, genetic variations in ion channel genes, or a combination of both factors [79]. The primary underlying cause can be attributed to ventricular arrhythmias, specifically ventricular tachycardia or ventricular fibrillation [80].

A deficiency of KCNE2 contributes to the development of atherosclerosis and diet-related sudden death by increasing plaque build-up and the formation of premature ventricular complexes, ultimately leading to sudden cardiac death [18]. Mouse models with a genetic deficiency in the *Kcne2* gene, were subjected to IR damage by coronary ligation. As a result, all of the mice experienced ventricular arrhythmias upon reperfusion. These conditions encompass ventricular tachycardia (VT), atrioventricular block (AVB), polymorphic ventricular tachycardia (PVT) or sustained ventricular tachycardia (SVT). However, the presence of the KCNE2 abnormality in mice undergoing RIPC processing results in reduced susceptibility to life-threatening ventricular arrhythmia [81]. Zhaoyang Hu et al. conducted a study to examine the specific capability of RIPC in fully preventing SCD and reducing the occurrence and intensity of ventricular arrhythmias and AV block in a mouse model of acquired LQTS, the *Kcne2*^−/−^ mouse line [81,82]. Activation of the AKT and ERK1/2 signaling pathways is essential for the cardioprotective effects of RIPC. These pathways play a crucial role in reducing ventricular arrhythmias, AVB and SCD.

The study conducted by Zhaoyang Hu et al. found that remote ischemic preconditioning (RIPC) of the limb significantly reduced the occurrence and intensity of all ventricular arrhythmias and completely prevented SCD compared to remote ischemic preconditioning of the liver. The reperfusion injury salvage kinase (RISK) signaling pathway, comprising protein kinase B (AKT) and extracellular signal-regulated kinase (ERK1/2) signaling molecules, and the survivor activating factor enhancement (SAFE) pathway, involving signal transducer and activator of transcription 3 (STAT-3), are inherent pro-survival signaling cascades in which proteins are phosphorylated and thus activated during remote ischemic preconditioning, thereby restricting the extent of infarcts [83,84]. Warsi et al. discovered that AKT activity enhances the activation of the Kv1.5 potassium channel protein on the cell surface [85]. In a mouse model, defects in the KCNE2 result in a reduced expression of Kv1.5 protein in the intercalated discs [16]. This would be expected to be antiarrhythmic in models such as the mouse defects in the KCNE2, in which Kv1.5 protein at the intercalated discs is reduced because KCNE2 is required for its efficient trafficking there. Furthermore, ERK1/2 is linked to the stimulation of potassium channels [86], LTCCs [87] and Na^+^/H^+^ exchanger [88], all of which have the potential to be targeted for antiarrhythmic effects.

#### 4.1.4. Atherosclerosis and Coronary Artery Disease

Soo Min Lee and colleagues have shown that genetic defects in certain subunits of ion channels can lead to the development of atherosclerosis. Other groups found that specific variations or single nucleotide polymorphisms (SNPs) within or near the KCNE2 gene were associated with a higher susceptibility to atherosclerosis, coronary artery disease (CAD) and early heart attack [89,90,91]. In addition, KCNE2 deficiency in mice promotes the development of atherosclerosis, as well as high-fat diet-dependent ventricular arrhythmia and sudden death [18].

#### 4.1.5. Myocardial Fibrosis

The absence of KCNE2 leads to an enlargement of the heart, mainly due to an increase in the size of the ventricular myocytes. KCNE2 deficiency in one-year-old mice has also been shown to lead to ventricular fibrosis [15], which can be effectively controlled by long-term use of angiotensin II receptor antagonists, as well as hypertrophic cardiomyopathy itself. The absence of KCNE2 led to an increase in serum angiotensin II and thus inactivation of GSK-3β to prevent cardiac hypertrophy [92]. GSK-3β in an active state is thought to enhance the activity of p53, which triggers the release of cytochrome c and causes destruction of mitochondria during the process of apoptosis [82]. On the other hand, deactivation of GSK-3β prevents the opening of the mitochondrial permeability transition pore (MPTP), which in turn halts the death of cardiomyocytes [93]. Therefore, it would be beneficial to investigate the cardioprotective effects of angiotensin II receptor antagonists on cardioprotection in *Kcne2*^−/−^ mice after fibrosis and ischemia–reperfusion injury (IRI).

### 4.2. Neurological Diseases

KCNE2 is highly expressed in the brain and interacts with several α-subunits of neuronal Kv channels, including Kv2.1, Kv4.2, Kv4.3, HCN1 and HCN2 [21]. KCNE2modulates the inactivation of KCNQ2 or KCNQ2/3 channels, hence influencing the M-type K^+^ currents in laboratory settings [22]. As mutations in both KCNQ2 and KCNQ3 have been linked to neonatal convulsions and epilepsy [94], any alterations in KCNE2 have the potential to influence the functionality of brain networks [21].

### 4.3. Aging

Aging refers to the gradual decline in physiological function that occurs over time. It results from many internal and external stresses related to the regulation of internal gene expression, metabolic processes and multiple signal transduction pathways [95]. However, the exact mechanisms remain unknown.

Oxidative damage caused by reactive oxygen species (ROS) is a major contributor to the functional decline and impairment of individual cells, which is among the biological hallmarks of aging. The data from the study by Eun-Ju Sohn et al. suggest that KCNE2 was upregulated in aged human dermal fibroblasts (HDFs). Silencing of KCNE2 reversed the gene expressions of EGR1 and p-ERK, and in turn decreased the expression levels of anti-oxidant enzymes such as superoxide dismutase and catalase in aged HDFs treated with anti-oxidant flavonoid quercetin [96]. The expression of KCNE2 declines with advancing age, resulting in the development of a number of illnesses. These conditions encompass T2DM, arrhythmia and hypothyroidism [14,24].

### 4.4. Achlorhydia, Gastric Hyperplasia and Neoplasia

KCNE2 and KCNQ1 together form the gastric potassium channels that are crucial for the process of gastric acidification. The absence of KCNE2 or KCNQ1 leads to increased acidity, excessive growth of the gastric mucosa, and the development of tumors.

The parietal potassium channel, consisting of KCNQ1 and KCNE2, is present in the mural cells. This channel allows the efflux of potassium ions, which in turn stimulates the secretion of gastric acid by the parietal potassium ATPase. Therefore, the removal of KCNQ1 or KCNE2 genes in mice interferes with the process of gastric acid secretion. Additional data suggest that KCNE2 is involved in the growth and proliferation of human gastric cancer cells, independent of its involvement in gastric acidity [97].

### 4.5. Hypothyroidism

Hypothyroidism is a common medical condition characterized by insufficient thyroid hormone. Without treatment, it can have significant adverse effects on many organ systems. Depending on the underlying pathology affecting the thyroid, pituitary gland, hypothalamus, or peripheral tissues, hypothyroidism is classified as primary, central, or peripheral. The most common type of hypothyroidism is acquired primary hypothyroidism, which can occur as a result of severe iodine deficiency. However, it is more commonly observed in areas with adequate iodine levels and persistent autoimmune thyroiditis [98].

The KCNQ1–KCNE2 potassium channel in thyroid epithelial cells has been shown to be essential for the full function of the Na^+^/I^−^ symporter (NIS), which serves as the primary pathway for iodine uptake in thyroid cells (Figure 4). Genetic disruption of the KCNQ1–KCNE2 genes in mice causes hypothyroidism, which in turn leads to cardiac hypertrophy, dwarfism, baldness and death during or shortly after birth [52].

### 4.6. Type 2 Diabetes Mellitus

The global incidence of T2DM is increasing, mainly as a result of lifestyle factors such as unhealthy diet and physical inactivity. T2DM is often associated with a broader condition known as the metabolic syndrome, which encompasses a number of comorbidities, including obesity, hypertension, non-alcoholic fatty liver disease (NAFLD), hypercholesterolemia and coronary artery disease [99].

The loss of KCNE2 also leads to extensive changes in the transcriptome of the pancreas, consistent with some features of T2DM, such as endoplasmic reticulum stress, inflammation, and excessive cell proliferation. Deletion of KCNE2 impairs the ability to secrete insulin and reduces insulin sensitivity. Deletion of KCNE2 causes a significant 8-fold reduction in β-cells insulin production in a laboratory setting. This deletion also reduces the peak outward K^+^ current of the β-cell membrane potential but shifts it toward more negative voltages and slows down the process of inactivation [19].

### 4.7. Lung Ischemia and Reperfusion Injury

Leng Zhou has documented that the KCNE2 is necessary for proper lung function in mice. The absence of KCNE2 leads to disturbances in blood gases, an increase in lung cell apoptosis and an increase in the levels of inflammatory mediators in both the fluid found in the lungs (bronchoalveolar lavage fluid) and the blood plasma. Significantly, the lack of KCNE2 also resulted in increased phosphorylation of several protective proteins in the reperfusion injury salvage kinase (RISK) signaling cascade. In addition, KCNE2 deficiency prevented the typical susceptibility and STAT-3 responses to pulmonary ischemia–reperfusion injury (IRI), thus increasing the levels of pulmonary IRI [100]. This further suggests that the absence of KCNE2 contributes to lung injury. Conversely, KCNE2 deletion has a different effect on cardiac IRI. It prepares the heart and reduces the severity of myocardial infarction caused by forced IRIs [101] (Figure 5).

### 4.8. Non-Alcoholic Fatty Liver Disease (NAFLD)

Non-alcoholic fatty liver disease (NAFLD) is a common liver disease in both developed and developing countries. NAFLD is defined by the abnormal accumulation of lipids in the liver, which can lead to the development of more serious conditions such as non-alcoholic steatohepatitis (NASH) and potentially fatal cirrhosis [102]. It typically occurs in people who are obese or diabetic, have unhealthy diets or lose weight rapidly. A number of genetic and epigenetic variables, as well as lifestyle choices and other external influences, also influence the incidence and severity of the diseases [103].

KCNE2 deficiency causes iron deficiency anemia, which can lead to dyslipidemia and NAFLD. Soo Min Lee discovered that mice lacking KCNE2 in their reproductive cells developed NAFLD as early as day 7 after birth [104].

### 4.9. Hepatocellular Carcinoma

In 2018, the global incidence of liver cancer was expected to be 841,000 new cases [105]. Hepatocellular carcinoma (HCC) accounts for the majority, specifically 75–85%, of liver cancer cases. Approximately 50% of liver cancer cases are expected in China [106]. Cancer cells exhibit unregulated proliferation, migration and invasion due to the abnormal expression of a number of several protein-coding genes and the inappropriate activation of signaling pathways [107].

Previous research has shown that KCNE2 is expressed at lower levels in gastric cancer. Artificially increasing KCNE2 expression suppresses cancer cell growth and cell cycle progression. Huamei Wei et al. showed that the expression of KCNE2 was reduced in hepatocellular carcinoma cell lines as compared to normal cell lines. In addition, the results indicate that decreased KCNE2 expression is associated with reduced overall survival in individuals with hepatocellular carcinoma (HCC). Suppression of KCNE2 expression was found to enhance the proliferation, migration and invasion of HCC cells. Furthermore, they found that miR-584-5p affected the behavior of HCC cells by specifically targeting KCNE2. The absence of miR-584-5p reduced the growth, migration, and invasion of HCC cells [108].

## 5. Conclusions and Perspectives

KCNE2 is a versatile protein that plays a role in several physiological processes and pathological conditions. Over the past few decades, many eminent researchers have made significant contributions to this topic, leading to the comprehensive understanding of the structure and biological role of KCNE2 in both health and diseases. KCNE2 does not have the ability to generate ion channels on its own. However, it acts as a complementary β-subunit to contribute to the regulation of α-subunits in other potassium channel proteins. At the same time, KCNE2 acts as a regulator, monitoring many aspects of the body to maintain regular physiological activities. KCNE2 has a crucial role in regulating the PPH level of gastric acid and is also important in the visual pathway. There is a growing body of evidence linking changes in KCNE2 expression to diseases factors. The diseases include tardive dyskinesia, sudden cardiac death, and T2DM. Remarkably, absence of KCNE2 can promote the development of atherosclerosis by causing plaques accumulation, the occurrence of premature ventricular complexes, and ultimately sudden cardiac death. Pre-existing remote ischemic preconditioning (in the limb or liver) significantly reduced the incidence and intensity of all ventricular arrhythmias and completely prevented sudden cardiac death. If validated in clinical trials, limb RIPC could serve as a non-invasive and non-pharmacological approach to reduce the occurrence of life-threatening ventricular arrhythmias resulting in SCDs associated with ischemia and/or channelopathies. Subsequent research will evaluate how the timing of RIPC in relation to the onset of ischemia and reperfusion affects its efficacy in the treatment of disease to demonstrate its practical application.

However, our knowledge of KCNE2 is still limited. There are still many unanswered questions that require further investigation. For instance, the relationship between KCNE2 and Kv1.3 remains challenging to elucidate. Mutations of KCNE2 have been identified in cases of neonatal epilepsy, indicating that KCNE2 may play a role in the inheritance of epilepsy. Nevertheless, the precise mechanism through which KCNE2 modulates neuronal excitability remains uncertain.

Moreover, it has been documented that KCNE2 activity is diminished in the ventricles of patients with heart failure, though the precise mechanism behind this phenomenon remains to be fully elucidated. To date, the findings has been conducted primarily on the rodent heart, which exhibits a rhythm ten times faster than that of humans, but is much smaller in size than the human heart. Consequently, the electrical activity is formed and propagated in a manner that differs between the two species. Additionally, it is noteworthy that rodent and human cardiac tissues contain a multitude of distinct ion channels, with the main difference thought to be in the Kv channel. A comprehensive study of KCNE2 function and its alterations in disease is essential to advance research in the field of heart failure. Furthermore, recent developments in research have identified modifications in KCNE2 in specific clinical specimens. In contrast, the available evidence on the regulation of KCNE2 expression is very limited. The definition of the promoter-regulated region of the KCNE2 gene remains inconclusive in all species. Our comprehension of the altered expression of KCNE2 under pathological conditions remains limited. Further research is required to address the outstanding questions in this area.

Consequently, further investigation into the function of KCNE2 and its involvement in disease pathogenesis will not only facilitate a great understanding of the molecular mechanisms underlying disease, but will also provide novel insights and avenues for the development of precise therapeutic strategies targeting KCNE2 and its associated ion channels.

## Figures and Tables

**Figure 1 cells-13-01409-f001:**
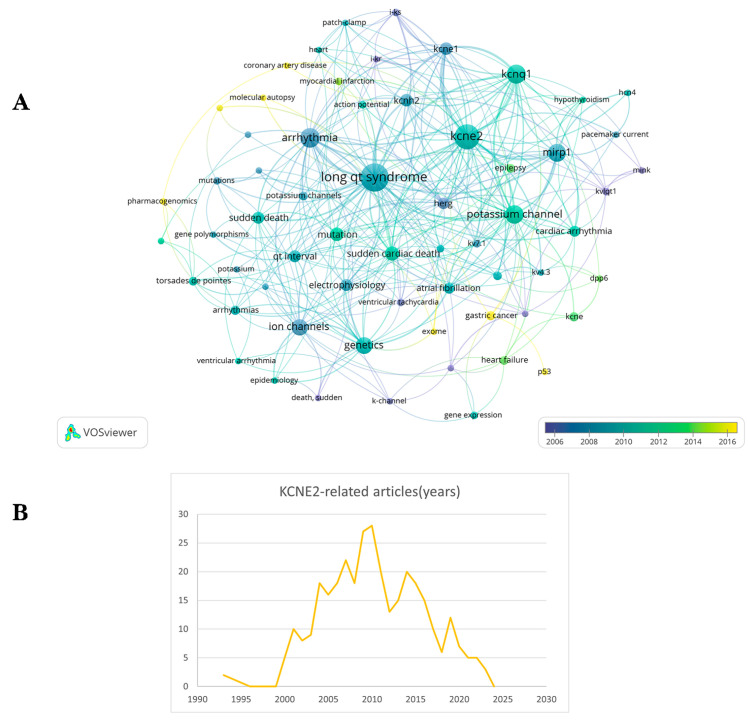
Statistics of KCNE2-related publications and a network visualization of keywords. (**A**) Bibliometric information obtained by VOS viewer showing frequent keywords in publications related to these KCNE2. (**B**) Statistics of publications related to KCNE2 since 1993.

**Figure 2 cells-13-01409-f002:**
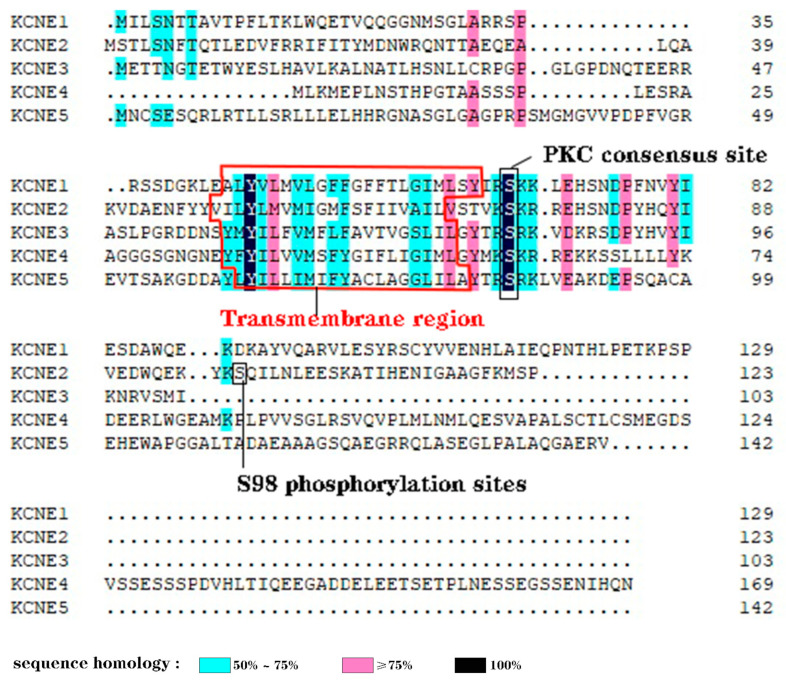
Human KCNE1–KCNE5 protein sequence alignments and homology level. Sequence images were generated using DNAMAN. sequence homology: light blue: 50%~75%; rose red: ≥75%; dark:100%.

**Figure 3 cells-13-01409-f003:**
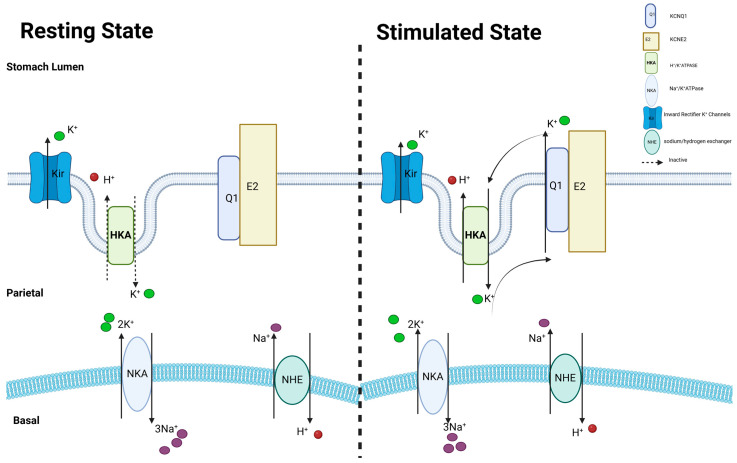
KCNE2 expressed in gastric epithelial cells. The arrangement of oxygenated glands in the stomach, with particular emphasis on the parietal cells. Gastric acidification in the middle region relies on the luminal K^+^-cycle pathway, composed of KCNQ1 (Q1) and KCNE2 (E2), and is necessary for the activity of the gastric H^+^/K^+^-ATPase.

**Figure 4 cells-13-01409-f004:**
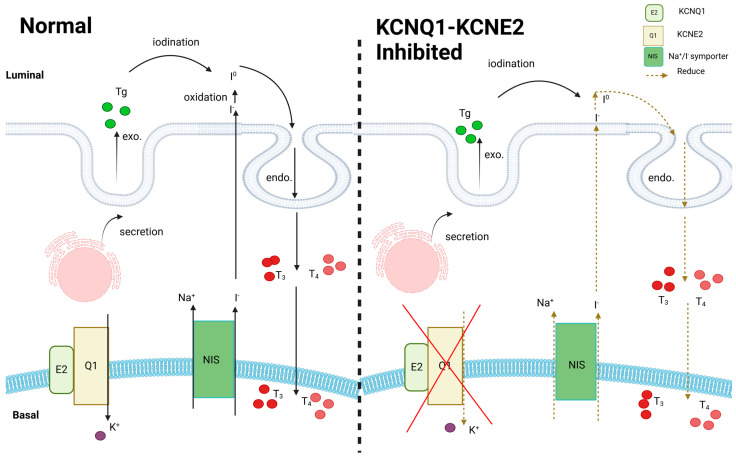
In a normal thyroid, iodine is taken up through the sodium–iodide symporter (NIS) for the iodination of thyroglobulin (Tg) and the production of thyroid hormones. The KCNQ1–KCNE2 complex is present on the basolateral membrane. Although pharmacological inhibition of the KCNQ1 gene and KCNE2 organization does not directly cause harm, it does result in reduced absorption and decreased production of thyroid hormones.

**Figure 5 cells-13-01409-f005:**
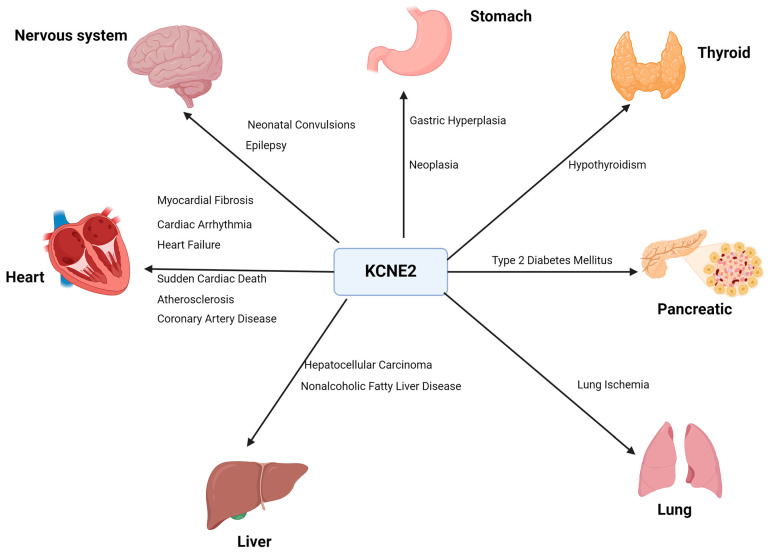
The association of KCNE2 with multi-organ pathophysiology and disease processes.

**Table 1 cells-13-01409-t001:** Pluripotent biological effects of KCNE2.

Channel	Biological Effects	Affected Tissue	References
LTCC	KCNE2 caused a positive shift in the activation voltage and a negative shift in the voltage inactivation of LTCC, and KCNE2 slowed down the recovery from inactivation and accelerates the deactivation of LTCC.	Heart	[34]
HCN channel	KCNE2 fasted activation and deactivation kinetics of channel, and increased current amplitude	Heart	[36,37,38]
HERG channel	S98-phosphorylated KCNE2 accelerates the degradation of HERG protein and to inhibit the amplitude of HERG current	Heart	[23]
I_to,fast_ channel	KCNE2 knockdown increased gating kinetics of I_to,fast_ in both neonatal and adult cardiomyocytes. Overexpression of KCNE2 reduced the activation and inactivation of I_to,fast_ in neonatal cardiomyocytes, whereas no effect on the gating properties of I_to,fast_ in adult cardiomyocytes	Heart	[10,16,40,41,42,43,44,45,46,47,48]
KCNQ1-KCNE2 channel	KCNQ1-KCNE2 channel could be activated by low extracellular pH and acts as a channel for potassium cycling in parietal cells. Furthermore, it is postulated to be essential in regulating the potassium flux, which in turn is crucial for the optimal production of thyroid hormones in thyrocytes and the production, secretion, and regulation of the CSF in CPe.	Stomach, thyroid and choroid plexus epithelium	[15,25,27,49,50,51]

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
