# Peer review of "The Multifunctional Role of KCNE2: From Cardiac Arrhythmia to Multisystem Disorders"

_cells, 2024, doi:10.3390/cells13171409_

Round 1

Reviewer 1 Report

Comments and Suggestions for Authors

The authors have written a timely update describing the physiology and pathobiology of a ubiquitously expressed and highly versatile gene, the KCNE2 (MiRP1) potassium channel ancillary subunit. they summarize the current knowledge of the gene well and describe the literature in a balanced manner. The review will be very useful for those seeking an update and a current understanding of the role of KCNE2, especially those interested in pursuing studies of this protein.

Specific points:

1) Abstract: it is worth noting that KCNE2 expression in the choroid plexus is as high as it is in the gastric epithelium.  This can also be mentioned on line 111.

2) Lines 115 and 121 - should be "parietal" not "mural" cells.

3) Line 43: put space between "as" and "MiRP1".

4) Figure 4 was referenced out of order, before figure 3.

5) Section 3.3: The first report of KCNE2 regulation of hERG and association with LQTS was Abbott et al 1999 (Cell).

6) Line 342: the report of plaque build-up in KCNE2 KO mice was by Lee et al J Mol 581 Cell Cardiol 2015, 87, 148-151. DOI: 10.1016/j.yjmcc.2015.08.013

7) Line 372: "They found" should be changed to "Other groups" as it was not Lee et al that found the human SNP association.

Comments on the Quality of English Language

minor edits needed to English

Author Response

We appreciate the positive evaluation of our study and constructive comments on it.

Comments 1: Abstract: it is worth noting that KCNE2 expression in the choroid plexus is as high as it is in the gastric epithelium.  This can also be mentioned on line 111.

Response 1: Thank you for pointing this out. We agree with this comment. We made the revision in the revised manuscript according to your suggestion. (Page 4 Lines 11)

Comments 2: Lines 115 and 121 - should be "parietal" not "mural" cells.

Response 2: Thank you for pointing this out. In the revised manuscript, we replaced the word “mural” with “parietal”. (Page 4, Lines 121)

Comments 3: Line 43: put space between "as" and "MiRP1"

Response 3: Thank you for pointing this out. We made the revision in the revised manuscript (Page 1 Lines 43)

Comments 4: Figure 4 was referenced out of order, before figure 3

Response 4: Thank you for pointing this out. We made the revision in the revised manuscript.

Comments 5: Section 3.3: The first report of KCNE2 regulation of hERG and association with LQTS was Abbott et al 1999 (Cell).

Response 5: Thanks for the comment. We add the reference according to your suggestion (Page 5, Lines 180,and Page 9, Lines 304).

Comments 6: Line 342: the report of plaque build-up in KCNE2 KO mice was by Lee et al J Mol 581 Cell Cardiol 2015, 87, 148-151. DOI: 10.1016/j.yjmcc.2015.08.013

Response 6: Thank you for pointing out this mistake. We made the revision in the revised manuscript (Page 10, Lines 364).

Comments 7:Line 372: "They found" should be changed to "Other groups" as it was not Lee et al that found the human SNP association.

Response 7:Thank you for pointing out this mistake. In the revised manuscript, we replaced the word “They found” with “Other groups”. (Page 11, Lines 395)

Reviewer 2 Report

Comments and Suggestions for Authors

This interesting review is focused on the molecular characteristics, molecular structure and tissue-specific distribution of KCNE2. The Authors discuss the 73 different functions of KCNE2, and particularly its roles in the development of various pathological conditions, such as cardiovascular, neurological, metabolic and multisystem disorders.

The main strength of this review is that it cover an innovative topic, not previously considered in literature.

Beyond the relevance of the topic, and the gap in the literature, the review appears rather complete. So, it is really of great interest for the scientific community.

References are appropriate and updated. The discussion and conclusion are supported by the references. The number of self-citations is limited.

Figures are well-done, easy to understand, and they help to interpret the data. 

Author Response

 We are grateful for your concise and exact summary and positive evaluation of our manuscript.

Reviewer 3 Report

Comments and Suggestions for Authors

KCNE2 is an ion channel regulatory subunit belonging to the KCNE family. It regulates several ion channel features shaping the physiological related consequences of the channels.

In this review, authors summarize the role of KCNE2 as an ancillary subunit of various ion channels. The work gives a general overview of KCNE2 in the context of the KCNE family, list the effects of KCNE2 on different ion channels and considers the different roles the protein plays in health and disease.

1.    Although the review provides accurate information on KCNE2 function, the manuscript can be throughout improved with some changes. Authors use old-fashioned nomenclature to refer to KCNE2 and other KCNE members. Authors should employ HUGO standard nomenclature through the text (KCNE instead of MinK or MiRPs, as well as CACNA or Cav for voltage-gated L-type calcium channels). The old nomenclature can be mentioned early in the introduction but should not be widely used to avoid confusions.

2.    The figures may be clearly improved bringing quality to the review. Figure 1 provides an overview of the bibliometrics of the KCNE family, which is not related neither to the text nor the aim of the article (focusing on KCNE2). If the authors want to compare the structure of the KCNE family members in Figure 2, they could add more information (e.g. transmembrane domains). In addition, they need to include a legend to the figure and pinpoint the specific KCNE2 residues mentioned in the text. Figures 3 and 4 are adaptations of figures previously published in other articles and lack of originality. This review should be improved if the authors presented figures of their own. The addition of either a figure or a table summarizing all the ion channels regulated by KCNE2 and its biophysical consequences would be of great help. Another figure schematizing the pathophysiology where KCNE2 is involved would further improve the manuscript.

3.    Section 3, focusing on KCNE2 regulation of specific ion channels lacks depth in some subsections (especially the 3.4. Ito current). Authors should add more information on KCNE2 regulation of the specific ion channels participating in Ito.

4.    Authors mention the KCNE2 regulation of Kv1.3. However, authors should be aware that this interaction is debatable because, in fact, it has not been observed throughout the literature. This point deserves clarification in a review.

5.    The conclusions and perspectives section could be improved if the authors listed which questions remain to be answered (or gaps in the literature) regarding the KCNE2 role.

6.    Although bibliography overall is adequate, in some sections the authors cite other reviews rather than original research papers (e.g. section 3.5.1, reference 25). In other sections, references do not correspond with the mentioned facts (e.g. sections 3.5.2, ref. 42). Authors should carefully examine the reference section.

7.    Minor editing issues are also noticed (e.g. lack or extra spaces between words).

Comments on the Quality of English Language

good enough

Author Response

We are grateful for your positive evaluation of our manuscript and the inspiring words. We have revised the manuscript according to your constructive suggestions which improved the quality of our manuscript.

Comments 1: Although the review provides accurate information on KCNE2 function, the manuscript can be throughout improved with some changes. Authors use old-fashioned nomenclature to refer to KCNE2 and other KCNE members. Authors should employ HUGO standard nomenclature through the text (KCNE instead of MinK or MiRPs, as well as CACNA or Cav for voltage-gated L-type calcium channels). The old nomenclature can be mentioned early in the introduction but should not be widely used to avoid confusions.

Response 1: Thanks for the constructive comment. According to your suggestion, we corrected the “MinK (KCNE1)” to “KCNE1”(Page5, Lines 165), and corrected the “MiRP1” to “KCNE2”(Page5, Lines 164-175).

The CACNA gene family, which encodes the alpha subunits of voltage-gated calcium channels complexes, is essential for forming functional calcium channels. We added the term LTCCs, the abbreviation for L-type Ca2+ channels, throughout the text. The CACNA1C gene encodes the calcium voltage-gated channel subunit alpha1C, also known as Cav1.2In the revised manuscript, we have conducted unification. The L-type Ca2+channel is designated as LTCC, and the alpha subunit is designated as cav1.2.

Comments 2: The figures may be clearly improved bringing quality to the review. Figure 1 provides an overview of the bibliometrics of the KCNE family, which is not related neither to the text nor the aim of the article (focusing on KCNE2). If the authors want to compare the structure of the KCNE family members in Figure 2, they could add more information (e.g. transmembrane domains). In addition, they need to include a legend to the figure and pinpoint the specific KCNE2 residues mentioned in the text. Figures 3 and 4 are adaptations of figures previously published in other articles and lack of originality. This review should be improved if the authors presented figures of their own. The addition of either a figure or a table summarizing all the ion channels regulated by KCNE2 and its biophysical consequences would be of great help. Another figure schematizing the pathophysiology where KCNE2 is involved would further improve the manuscript.

Response 2: Thanks for the constructive comment. According to your suggestion,

we redraw Figure 1 with a particular focus on KCNE2.

In Figure 2, further details have been included, such as information pertaining to transmembrane domains and the S98 phosphorylation site.

In accordance with your recommendation, we have revised Figures 3. Given the limited number of studies, we have simplified Figure 4 by incorporating relevant information from relevant literature, thereby enhancing its accessibility.

A new figure, Figure 5, has been included in the revised manuscript, which illustrates the association between KCNE2 and multi-organ pathophysiology and disease processes. processes.In accordance with your constructive suggestions, we have incorporated a table summarizing all the ion channels regulated by KCNE2 and its biophysical consequences, which has led to an improvement in the quality of our manuscript.

Comments 3: Section 3, focusing on KCNE2 regulation of specific ion channels lacks depth in some subsections (especially the 3.4. Ito current). Authors should add more information on KCNE2 regulation of the specific ion channels participating in Ito.

Response 3: We appreciate the constructive comment. We added paragraphs discussing this issue, as” The transient outward current Ito starts the early repolarization phase of the heart action potential and helps decide the repolarization phase and excitation-contraction coupling by controlling Ca2+ and other K+ currents41, 42. Ito is made up of two main parts: Ito,fast and Ito, slow. The molecules that go with Ito, fast are Kv4.2 and/or Kv4.3, and the molecules that go with Ito, slow are Kv1.443-46. KCNE2 joins with both Kv4.2 and Kv1.5 in the ventricles of adult mice, making their currents stronger.

Erich Wettwer et al. suggested that KCNE2 might be a key part of the native Ito channel complex, at least in human epicardial cardiomyocytes10. When KCNE2 was co-expressed with Kv4.3 in Xenopus oocytes, it slowed down the process of Ito activation and deactivation by a lot, and it changed the voltage dependence of activation to a positive membrane potential10. There was a difference, though, between KCNE2 and what they expected in the CHO cell translation system. Also, they discovered that KCNE2 is unique because its co-expression best replicates the unique "overshoot" seen during the return of Ito channel inactivation in the human left ventricular epicardium47.” (Page 6, Line 191-204)

Comments 4:  Authors mention the KCNE2 regulation of Kv1.3. However, authors should be aware that this interaction is debatable because, in fact, it has not been observed throughout the literature. This point deserves clarification in a review.

Response 4: We appreciate the constructive comment.  We added a paragraph discussing this issue, as “KCNE2 also forms apical potassium channels at the CPe with Kv1.3 (KCNA3), which is believed to contribute to apical K+ efflux 28. While KCNQ1 loses its voltage-dependence when co-expressed with KCNE2, Kv1.3-KCNE2 channels maintain their voltage sensitivity, but display a reduced overall current strength in comparison to the Kv1.3 channels. The specific reason for this decrease in current is still unknown28. In a separate study, Sole L et al observed that human KCNE2 did not exert any regulatory effect on the function of rat Kv1.3 in HEK cells55. (Page 9, Line 246-253)

Comments 5: The conclusions and perspectives section could be improved if the authors listed which questions remain to be answered (or gaps in the literature) regarding the KCNE2 role.

Response 5: We appreciate the constructive comment. We added paragraphs, as” It remains uncertain how modifying KCNE2's function within the CPe membrane could potentially alter neuronal excitability. An investigation has been conducted to ascertain the potential involvement of KCNE2 in CPe, due to its elevated expression levels in this region compared to other brain areas. Nevertheless, the observation that KCNE2 regulates KCNQ2-KCNQ3 and HCN (pacemaker) channels in vitro, with the possibility of these channels being expressed in a shared neuronal population, indicates that KCNE2 may also exert a direct influence on specific neurons.

As our understanding of KCNE2 has deepened, we have not only identified the intricate and varied functions of this protein, but also established a clear link between its expression and the physiological and pathological states of multiple organs and systems. Abnormal KCNE2 expression or dysfunction has been shown to result in a range of pathological changes, which may contribute to the development of conditions such as cardiac arrhythmia, epilepsy, and diabetes mellitus. This discovery significantly enhances our comprehension of the function of KCNE2 in disease pathogenesis and furnishes a crucial scientific foundation for our understanding of the mechanisms un-derlying the formation of ion channel-related diseases. This not only facilitates the elucidation of the molecular mechanisms underlying disease pathogenesis, but also offers novel insights and avenues for the development of precision therapeutic strate-gies targeting KCNE2 and associated ion channels. It is therefore evident that further research into the function of KCNE2 and its role in disease is of great theoretical and practical significance for the future clinical diagnosis, treatment and prevention of re-lated diseases.” (Page 15, Line 550-570)

Comments 6: Although bibliography overall is adequate, in some sections the authors cite other reviews rather than original research papers (e.g. section 3.5.1, reference 25). In other sections, references do not correspond with the mentioned facts (e.g. sections 3.5.2, ref. 42). Authors should carefully examine the reference section.

Response 6: Thank you for pointing this out. We read through the entire text and corrected incorrect references.

Comments 7: Minor editing issues are also noticed (e.g. lack or extra spaces between words).

Response 7: Thank you for pointing this out. We read through the entire text carefully and made corrections for editing issues.

Reviewer 4 Report

Comments and Suggestions for Authors

This manuscript revolves around KCNE2 and its role in physiological and pathophysiological processes. The authors discussed the structural and functional properties of KCNE2. They also summarized KCNE2 expression in different kinds of cells and tissues and its modulation of various ion channels. This review is thorough, which would attract a wide range of readers. However, I have several concerns and suggestions below.

1. Line 151, the abbreviation for L-type Ca2+ channels is normally LTCCs.

2. Line 182, S98 could be added to Figure 2.

3. Line 195, What does “gating properties of adult cardiomyocytes” mean?

4. Figure 3, Ikr is not Inward Rectifier K channels, Kir is.

5. Some grammar errors, such as Line 395,

Comments on the Quality of English Language

The authors need to check for some grammar errors. 

Author Response

Thanks for your concise and exact summary and positive evaluation of our study.

Comments 1: Line 151, the abbreviation for L-type Ca2+ channels is normally LTCCs.

Response 1: Thank you for pointing this out. In the revised manuscript, we corrected the “LCCs” to “LTCCs”(Page 5, Lines 151)

Comments 2: Line 182, S98 could be added to Figure 2.

Response 2: Thank you for your critical comment. We added the S98 phosphorylation sites in Figure 2 according to your suggestion.

Comments 3: Line 195, What does “gating properties of adult cardiomyocytes” mean?

Response 3: Thank you for pointing out this mistake. We missed “Ito in” in the original manuscript, and added it in the revised manuscript. (Page 6, Lines 209)

Comments 4: Figure 3, Ikr is not Inward Rectifier K channels, Kir is.

Response 4: Thank you for pointing out this mistake. We made the revision in the revised manuscript.

Comments 5: Some grammar errors, such as Line 395

Response 5: Thank you for pointing out this mistake. The revised sentence is” As mutations in both KCNQ2 and KCNQ3 have been linked to neonatal convulsions and epilepsy82, any alterations in KCNE2 have the potential to influence the functionality of brain networks22.” (Page 12, Lines 418-420). We read through the entire text carefully and made corrections for grammatical errors.

Round 2

Reviewer 3 Report

Comments and Suggestions for Authors

We thank the authors because all points have been mostly adequately addressed. However, some concerns are still pending.

- Table 1: authors should specify the effects on the different subunits (e.g. affects the opening: in which direction? "affects the functionality" how?)
- New Ito paragraph (line 190-204): vocabulary should be more precise ("the molecules that go with Ito are...", "currents stronger" lines 194-196).
- Additional Conclusions paragraph: seems redundant and not including new perspectives nor summarizing concepts in a new way.
- Figure 2: add legend to the figure for the coloring of the residues.
- lines 118-127 and 246-251: Answering previous reviewer’s concerns, these paragraphs are now almost identical, this needs to be fixed.

- Former comment 4 (previous reviewer’s concerns):  Authors mention the KCNE2 regulation of Kv1.3. However, authors should be aware that this interaction is debatable because, in fact, it has not been observed throughout the literature. This point deserves clarification in a review.

Author’s Response 4: We appreciate the constructive comment.  We added a paragraph discussing this issue, as “KCNE2 also forms apical potassium channels at the CPe with Kv1.3 (KCNA3), which is believed to contribute to apical K+ efflux 28. While KCNQ1 loses its voltage-dependence when co-expressed with KCNE2, Kv1.3-KCNE2 channels maintain their voltage sensitivity, but display a reduced overall current strength in comparison to the Kv1.3 channels. The specific reason for this decrease in current is still unknown28. In a separate study, Sole L et al observed that human KCNE2 did not exert any regulatory effect on the function of rat Kv1.3 in HEK cells55. (Page 9, Line 246-253)

Author’s reply tries to elaborate a more open debate. However, they only/subtly mention Sole et al., (2009). The debate goes beyond that work. In fact, neither Abbott et al., Cell 97 (1999) 175-187 nor Grunnet et al., Biophys J 85 (2003) 1525-1537 find effects in Xenopus oocytes. Therefore, these findings should be mentioned to confront the unique, yet uncertain, positive interaction in CP by Roepke et al., FASEB J. 25(2011) 4264-4273. In fact, Abbott's work (taken as reference) himself published negative (1999) and positive (2011) results. In a review this debate should be mentioned in more detail because could be reference for readers.

Author Response

Response: We appreciate your constructive comments on it. We revised the manuscript according to your suggestions which improved the quality of our manuscript.

Comments 1: Table 1: authors should specify the effects on the different subunits (e.g. affects the opening: in which direction? " affects the functionality" how?)

Response 1: Thank you for your constructive comments. We made the revision in the revised manuscript

Comments 2: New Ito paragraph (line 190-204): vocabulary should be more precise ("the molecules that go with Ito are...", "currents stronger" lines 194-196).

Response 2: Thank you for pointing this out. We made the revision in the revised manuscript (Page 5, line 184- 201).

Comments 3: Additional Conclusions paragraph: seems redundant and not including new perspectives nor summarizing concepts in a new way.

Response 3: Thank you for pointing this out. We rewrote the Conclusions paragraph in the revised manuscript. (Page 15, Line 564-577. Page 16, Line 578-584)

Comments 4: Figure 2: add legend to the figure for the coloring of the residues.

Response 4: Thank you for pointing this out. We made the revision in the revised manuscript.

Comments 5: lines 118-127 and 246-251: Answering previous reviewer’s concerns, these paragraphs are now almost identical, this needs to be fixed.

Response 5: Thank you for your constructive comments. We made the revision in the revised manuscript.(Page 4, Line 117-120)

Comments 6:Former comment 4 (previous reviewer’s concerns):  Authors mention the KCNE2 regulation of Kv1.3. However, authors should be aware that this interaction is debatable because, in fact, it has not been observed throughout the literature. This point deserves clarification in a review. 

Author’s Response 4: We appreciate the constructive comment.  We added a paragraph discussing this issue, as “KCNE2 also forms apical potassium channels at the CPe with Kv1.3 (KCNA3), which is believed to contribute to apical K+ efflux 28. While KCNQ1 loses its voltage-dependence when co-expressed with KCNE2, Kv1.3-KCNE2 channels maintain their voltage sensitivity, but display a reduced overall current strength in comparison to the Kv1.3 channels. The specific reason for this decrease in current is still unknown28. In a separate study, Sole L et al observed that human KCNE2 did not exert any regulatory effect on the function of rat Kv1.3 in HEK cells55. (Page 9, Line 246-253)

Author’s reply tries to elaborate a more open debate. However, they only/subtly mention Sole et al., (2009). The debate goes beyond that work. In fact, neither Abbott et al., Cell 97 (1999) 175-187 nor Grunnet et al., Biophys J 85 (2003) 1525-1537 find effects in Xenopus oocytes. Therefore, these findings should be mentioned to confront the unique, yet uncertain, positive interaction in CP by Roepke et al., FASEB J. 25(2011) 4264-4273. In fact, Abbott's work (taken as reference) himself published negative (1999) and positive (2011) results. In a review this debate should be mentioned in more detail because could be reference for readers.

Response 6: We appreciate the constructive comment. We added a paragraph discussing this debate,as“However, our knowledge of KCNE2 is still limited. There are still many unan-swered questions that require further investigation. For instance, the relationship be-tween KCNE2 and Kv1.3 remains challenging to elucidate. Mutations of KCNE2 have been identified in cases of neonatal epilepsy, indicating that KCNE2 may play a role in the inheritance of epilepsy. Nevertheless, the precise mechanism through which KCNE2 modulates neuronal excitability remains uncertain.

Moreover, it has been documented that KCNE2 activity is diminished in the ven-tricles of patients with heart failure, though the precise mechanism behind this phe-nomenon remains to be fully elucidated. To date, the findings has been conducted pri-marily on the rodent heart, which exhibits a rhythm ten times faster than that of hu-mans, but is much smaller in size than the human heart. Consequently, the electrical activity is formed and propagated in a manner that differs between the two species. Additionally, it is noteworthy that rodent and human cardiac tissues contain a multi-tude of distinct ion channels, with the main difference thought to be in the Kv channel. A comprehensive study of KCNE2 function and its alterations in disease is essential to advance research in the field of heart failure. Furthermore, recent developments in re-search have identified modifications in KCNE2 in specific clinical specimens. In con-trast, the available evidence on the regulation of KCNE2 expression is very limited. The definition of the promoter-regulated region of the KCNE2 gene remains inconclusive in all species. Our comprehension of the altered expression of KCNE2 under pathological conditions remains limited. Further research is required to address the outstanding questions in this area.

Consequently, further investigation into the function of KCNE2 and its involve-ment in disease pathogenesis will not only facilitate a great understanding of the mo-lecular mechanisms underlying disease, but will also provide novel insights and ave-nues for the development of precise therapeutic strategies targeting KCNE2 and its as-sociated ion channels.”. (Page 14, Line 554-555; Page 15, Line 556-579)

Round 3

Reviewer 3 Report

Comments and Suggestions for Authors

No further concerns